# Cerebral Folate Deficiency Syndrome: Early Diagnosis, Intervention and Treatment Strategies

**DOI:** 10.3390/nu14153096

**Published:** 2022-07-28

**Authors:** Vincent Th. Ramaekers, Edward V. Quadros

**Affiliations:** 1Autism Reference Center, University of Liege, 4000 Liege, Belgium; 2SUNY Downstate Medical Center, Brooklyn, NY 11203, USA; edward.quadros@downstate.edu

**Keywords:** folate transport, folate receptor autoantibodies, cerebral folate deficiency, inborn errors of folate metabolism, prenatal folate deficiency

## Abstract

Cerebral folate deficiency syndrome (CFDS) is defined as any neuropsychiatric or developmental disorder characterized by decreased CSF folate levels in the presence of normal folate status outside the nervous system. The specific clinical profile appears to be largely determined by the presence or absence of intrauterine folate deficiency as well as postnatal age at which cerebral folate deficiency occurs. The primary cause of CFDS is identified as the presence of serum folate receptor-alpha (FRα) autoantibodies impairing folate transport across the choroid plexus to the brain whereas, in a minority of cases, mitochondrial disorders, inborn errors of metabolism and loss of function mutations of the FRα (*FOLR1*) gene are identified. Early recognition and diagnosis of CFDS and prompt intervention is important to improve prognosis with successful outcomes. In this article we focus on FRα autoimmunity and its different age-dependent clinical syndromes, the diagnostic criteria, and treatments to be considered, including prevention strategies in this at-risk population.

## 1. Introduction

Folic acid is a water-soluble vitamin B9 and represents the oxidized form of folates, a family of more than 25 structurally related and interconvertible compounds. The common backbone of folates is a core structure of an unreduced 2-amino-4-hydroxypteridine molecule linked through a methylene carbon at the C6 position to a para-aminobenzoyl group, which is attached to a mono-glutamate or poly-glutamate chain. Reduction in the pteridine ring at the 5,6,7,8 carbon positions results in reduced folate forms, which participate in intermediary metabolism. Between the N^5^-position of the pteridine group and the N^10^-position at the para-aminobenzoyl group, a one-carbon group at different oxidation states in the form of a methyl-, methylene-, methenyl- or formyl-group is attached. The physiologically active tetrahydrofolate is reduced at the 5,6,7,8 positions of the pteridine ring [1] (see Figure 1).

Folic acid and folates are a group of interconvertible vitamin B9 forms, which are essential for fertility, embryonic/fetal development and postnatal neurodevelopment, and play multiple roles in intermediary metabolism and epigenetic imprinting, including regulation of gene expression (see Figure 2). It should be noted that vitamin B2, B6 and B12 serve as co-factors for various enzymes interconverting different folate forms, such as the folate dependent homocysteine degradation, the re-methylation cycle, the methionine cycle (methionine and SAM production), the transsulfuration cycle and purine and thymidylate synthesis. Thus, an adequate pool of these co-factors (vitamin B2, B6, B12) and normal enzyme functioning is required to interconvert different folate forms since each specific folate form participates in the methionine, remethylation and transsulfuration cycles, and thymidine and purine production, resulting in folate dependent cell proliferation, homocysteine catabolism, multiple SAM-mediated methyl-transfer reactions, neurotransmitter synthesis and epigenetic mechanisms [2].

Folate absorption occurs primarily in the jejunum and is mediated by the proton coupled folate transporter (*PCFT*) expressed on the intestinal enterocytes, while a minor proportion is absorbed through the reduced folate carrier (*RFC1*). In the intestinal epithelial cells, folic acid is reduced by dihydrofolate reductase (DHFR) to its tetrahydrofolate (THF) form which then goes into the methylene-THF pool and is ultimately converted to N^5^-methyl-tetrahydrofolate (MTHF). At higher doses, some folic acid or folinic acid is transported into the circulation unconverted, but further conversion to MTHF can occur in the liver and other tissues [3]. Folates can be stored intracellularly by attachment of poly-glutamate chains which can range from 5 to 8 glutamic acid residues. The main circulating folate form in plasma is MTHF, which represents the reduced folate form able to pass the choroid plexus. Recycling of tetrahydrofolate to MTHF requires the addition of a one-carbon group through several enzymatic steps, where the carbon unit is provided by the so-called single-carbon exchange pool of the amino-acids histidine, serine, and glycine.

MTHF, the main circulating folate form, is transported across the blood–brain barrier at the choroid plexus into the spinal fluid compartment through the combined action of the GPI-anchored membrane folate receptor-α (FRα) and *PCFT* utilizing an endocytic vesicular transport process. The choroid plexus is the main site for physiological folate transport into the CNS. Subsequently, MTHF can cross the ependymal barrier to enter the brain parenchyma where it will finally be incorporated into neurons and other cells [4].

A similar FRα-mediated transport process appears to operate at the placental barrier for the passage of folates to the fetus. Other sites where FRα-mediated endocytosis plays a role for physiological folate transport are the male and female gonads, and the thyroid gland in utero and during the first months after birth [5,6,7].

## 2. Hereditary Folate Malabsorption and Cerebral Folate Deficiency (CFD)

In 1931, Lucy Wills reported an unusual form of macrocytic anemia in India, which could be treated by Marmite, a yeast extract suggesting, that it was the consequence of a nutritional deficiency [8]. Later, the active substance provided by yeast extract was identified as folic acid [9].

Intrauterine folate deficiency during pregnancy has been associated with neural tube defects in the fetus and predisposition to developmental deficits, including autism [10,11,12,13]. Severe systemic folate deficiency has been associated with macrocytic anemia and additional pancytopenia, immune dysfunction, and neurologic disorder. The best example demonstrating the consequences of systemic folate deficiency is hereditary folate malabsorption due to gene defects of the proton-coupled folate transporter (*PCFT*), localized in the small intestine and at the choroid plexus, which leads to impaired intestinal folate absorption and transport across the choroid plexus to the brain [14]. This rare genetic condition has autosomal recessive inheritance with failure to thrive, oral ulcers, poor feeding, diarrhea, folate malabsorption, increased susceptibility to infections, folate-responsive megaloblastic anemia, thrombocytopenia, neutropenia, and low immunoglobulins. Neurological manifestations of the CNS include irritability, head lag and hypotonia, mental retardation, ataxia, seizures, athetosis, dyskinesias and peripheral neuropathy. CT-brain scan has shown calcifications in basal ganglia. Laboratory findings show decreased serum and cerebrospinal fluid (CSF) folate, low plasma methionine and increased urinary excretion of formiminoglutamic acid (FIGLU). Early diagnosis and folinic acid treatment with IM injections can avoid neurologic sequelae [15,16].

In 2004, we introduced the umbrella term cerebral folate deficiency (CFD) to define any neuropsychiatric or neurodevelopmental disorder associated with low MTHF concentration in CSF in the presence of normal folate, vitamin B12 and homocysteine outside the nervous system [1]. Thus, for an accurate diagnosis of CFD, it is necessary to measure MTHF in spinal fluid because blood folate and metabolite status do not reflect CNS folate deficiency.

There are five potential mechanisms underlying CFD (see Table 1):I.Reduced transport of folates across the blood–brain barrier into CSF and across the ependymal into the brain.II.Reduced folate storage and release from the intracellular folyl-polyglutamate pool.III.Increased utilization and consumption of reduced folates within the nervous system leading to depletion of the folate poolIV.Increased catabolism of reduced folates within the nervous systemV.Metabolic conditions affecting folate metabolism within the nervous system

Table 2 provides a list of hitherto identified or postulated mechanisms contributing to systemic folate depletion and CFD syndrome (CFDS).

The first described clinical syndrome of CFD, called infantile-onset CFD, resembles hereditary folate malabsorption with respect to its neurological features, but shows a less severe clinical profile without hematological, intestinal or immunological abnormalities as seen in hereditary folate malabsorption, since the latter represents a systemic folate deficiency state [17,18].

Children with infantile CFDS were recognized at onset between 4–6 months to manifest sleeping problems, unrest and agitation, and to develop the full clinical phenotype over the next two years. After the first symptoms and signs of agitation and insomnia have manifested, there is a deceleration of head growth, psychomotor retardation with hypotonia and ataxia, occurrence of distal pyramidal signs, and in one third of patients, development of dyskinesias (chorea-athetosis, dystonia, hemi-ballism) and/or epileptic seizures. In untreated patients, visual loss develops from the age of three and bilateral hearing loss from the age of six years. In contrast to hereditary folate malabsorption, the brain scans in infantile CFD syndrome lack basal ganglia calcifications. MRI brain scans among half of untreated CFD cases show delayed myelination with demyelinated subcortical and periventricular areas, cerebral and cerebellar atrophy. Laboratory results are normal, including hematology, folate, B_12_, homocysteine, immunoglobulins, amino acids and screening for inborn errors of metabolism [1,17,18,19].

## 3. Causes of Systemic Folate Deficiency

Worldwide systemic folate deficiency occurs frequently and is due to malnutrition [20]. Malnutrition can also be associated with poor feeding in alcoholism and occurs among many individuals suffering from psychiatric diseases with eating disorders [21,22,23,24]. Malabsorption with reduced folate absorption in the jejunum can occur in gluten enteropathy, chronic intestinal inflammatory disorders such as Chron´s disease and recurrent chronic intestinal infections and lambliasis [25,26,27].

Antifolate drugs used in cancer chemotherapy, such as methotrexate (MTX), 5 fluoro-Uracil (5FU) and Pemetrexed, directly interfere with folate utilization [28]. Other drugs deranging folate metabolism and cellular folate uptake are the tuberculostatic drug isoniazide, sulphonamides (analogs of para-aminobenzoic acid, interfering with THF synthesis in sensitive bacteria), anticonvulsant drugs such as valproic acid, phenytoine, and carbamazepine (toxic drug metabolites and formed radical oxygen species interfering with cellular folate uptake), and carbidopa (inhibiting aromatic amino acid decarboxylase with consequent SAM and 5MTHF overconsumption) [29,30,31,32,33].

Congenital folate malabsorption is a rare hereditary autosomal recessive condition due to *PCFT* gene mutations, with characteristic clinical features mentioned above [14,15,16].

Inborn errors of metabolism include hereditary methylene-THF reductase (MTHFR) deficiency due to rare mutations with very low MTHFR enzyme activity or due to common mutations with about 50–60 % reduction in MTHFR enzyme activity, which can present without symptoms but also affect folate inter conversions in some instances since its reaction product MTHF in plasma will diminish consequently and predispose to lower available MTHF levels to cross the blood–brain barrier. The inborn error of metabolism associated with very low MTHFR enzyme activity presents with hyper homocysteinemia and homocystinuria, developmental delay, eye disorders, thrombosis and osteoporosis [34]. The common homozygous (C677T) mutations or compound heterozygous (C677T and A1298C) mutations are expected to lead to even lower MTHFR enzyme activity in the presence of low levels of its co-factor riboflavin [35,36].

Dihydrofolate reductase (DHFR) deficiency is a rare autosomal recessively transmitted disorder with megaloblastic anemia, reduced red blood cell folate and low CSF MTHF. There is a wide range of clinical neurologic manifestations reminiscent of infantile CFD [37,38].

Glutamate formiminotransferase deficiency is a hereditary autosomal recessive condition due to defective delivery of histidine derived one-carbon groups to tetrahydrofolate (THF), with a variable clinical picture showing, in some instances, manifestations of megaloblastic anemia, physical growth retardation and mental sub-normality. Metabolic urinary screening reveals formiminoglutamic aciduria [39].

The recently described autosomal recessive condition methylenetetrahydrofolate dehydrogenase-1 (*MTHFD1)* deficiency presents with megaloblastic anaemia, variable hyper homocysteinemia, atypical haemolytic uremic syndrome, combined immune deficiency and, in some cases, mental retardation and seizures. It is caused by mutations in the *MTHFD1* gene resulting in deficiency of a trifunctional enzyme containing 5,10-methylenetetrahydrofolate dehydrogenase, 5,10-methenyltetrahydrofolate cyclohydrolase and 10-formyltetrahydrofolate synthetase activities [40].

## 4. Causes of Cerebral Folate Deficiency (CFD)

In children suffering from infantile-onset CFDS, the most prevalent cause now appears to be the presence of serum autoantibodies directed against the choroid plexus membrane-attached GPI-anchored FRα (FRαAb), with consequent impaired MTHF binding and passage across the choroid plexus to the spinal fluid compartment and to the brain [41]. Two types of FRαAb can be distinguished: the so-called blocking autoantibodies which bind at or near the folate binding site of FRα, and the so-called binding FRα autoantibodies which bind to the FRα at a site distant from the folate binding site. Both types of antibodies can produce complement-mediated inflammation followed by destruction of the FRα-autoantibody complex [42], which results in a reduction in overall folate transport to the brain. In other CFD syndromes presenting at a later age in childhood, adolescence or adulthood, serum FRα autoimmunity is also present in many patients [2,3,43,44,45,46].

Our studies indicate that the human FRα protein antigen shows around 90% amino acid sequence homology to FRα antigen present in all animal-derived milk and milk products. Among genetically susceptible patients, the ingestion of cow´s milk after birth predisposes the intestinal immune system to produce antibodies against the soluble FRα antigen present in cow’s milk which can enter the circulation and cross-react with the FRα antigen attached to the choroid plexus, thyroid and gonads. An animal-derived milk-free diet can down-regulate the production of these cross-reacting FRα autoantibodies over a period of three months, whereas re-exposure will provoke the production of these antibodies again [47].

Moreover, an inverse correlation exists between the titer of FRα autoantibodies and CSF MTHF levels. Another observation among CFD patients was that FRα autoantibody titers may fluctuate in cycles over 5–7 weeks, where on one occasion FRα antibodies show negative or low titer and subsequently show high FRαAb titer a few weeks later [47].

Another group of causes underlying CFD syndrome are mitochondrial defects associated with nuclear- or mitochondrial-encoded DNA mutations at the basis of mitochondrial encephalopathies, Kearns–Sayre syndrome (large defects of mitochondrial DNA [48,49]) and Alper´s disease (mitochondrial DNA depletion due to failure of DNA replication by the DNA polymerase-gamma enzyme, encoded by the *POLG* gene; [50]). Since these mitochondrial disorders disturb oxidative phosphorylation and ATP production, the ATP-dependent active folate transport by the choroid plexus and the intracellular folate storage by ATP-requiring folylpolyglutamate synthetase will diminish [51].

A number of cases with very low CSF MTHF levels have been due to FRα protein encoding autosomal recessive *FOLR1* mutations resulting in functional loss of FRα. Clinical features are reminiscent of FRα autoantibody mediated infantile-onset CFD syndrome, although signs and symptoms are more pronounced [52]. Evaluation of 48 sporadic CFD patients as a validation cohort, identified four rare de novo variants of the capicua transcriptional repressor gene (CIC) that are putative deleterious mutations. Functional analysis indicates that CIC binds to an octameric sequence in the promoter region of folate transport genes: *FOLR1*, *PCFT* and *RFC1. CIC* loss of function variants can contribute to the genetic etiology of CFD through downregulating *FOLR1* expression [53].

There are several inborn errors affecting folate metabolism and MTHF homeostasis in the CNS. A one-carbon pool deficiency in the CNS is the autosomal recessive inherited 3-Phosphoglycerate dehydrogenase deficiency which is a disorder of serine synthesis leading to a low serine pool. As serine participates together with histidine and glycine as one-carbon donors to folates, it will affect one-carbon unit attachments to tetrahydrofolate in the CNS by the B6-dependent enzyme serine hydroxymethyl transferase [54].

Another rare autosomal recessive inborn error is methenyl-THF synthetase deficiency, leading to defective conversion of 5-formyl-THF to 5–10-methenyl-THF, resulting in elevated intracellular 5-formyl-THF levels and reduced MTHF spinal fluid levels within the low normal range. Treatment with folinic acid (5-formyl-THF) aggravates clinical and biochemical features of this CFD syndrome [55].

In autosomal recessive dihydropteridine reductase (DHPR) deficiency, the substrate dihydrobiopterin accumulates because it cannot be reduced to tetrahydrobiopterin, the active cofactor of aromatic amino acid hydroxylases, NO-synthase and phenylalanine hydroxylase, resulting in deficient production of CNS monoamines, nitric oxide (NO) with simultaneous elevation of plasma phenylalanine (phenylketonuria) [56]. In the brain, the reduction of dihydrofolate to THF is mediated solely by DHPR, because the dihydrofolate reductase (DHFR) enzyme is extremely low in the human brain [57]. This explains why, in patients with DHPR deficiency, the consequent reduction in the THF and MTHF pool in the brain can lead to CFD and renders treatment, with reduced folate analogues necessary.

Autosomal recessively inherited aromatic L-amino acid decarboxylase deficiency leads to deficient decarboxylation of L-dopa and 5-hydroxytryptophan to dopamine and serotonin, respectively. The accumulated L-dopa in the CNS will be converted to 3-O-methyl-dopa using the enzyme catecholamine methyltransferase (*COMT*), which uses the universally activated methyl-donor S-adenosyl-methionine (SAM). Consequently, this catabolic pathway of L-dopa utilizes high amounts of SAM, which depends on adequate MTHF reserves [33]. Thus, there is an overconsumption of MTHF and its activated methyl-donor SAM (see methionine cycle pathway). Aromatic amino acid decarboxylase is a B6-dependent enzyme and can be inhibited by carbidopa or benserazide, used as peripheral decarboxylase inhibitors in treatment of Parkinson’s disease. Thus, it is very likely that the combined L-dopa/carbidopa treatment for Parkinson´s disease will result in overconsumption of SAM and MTHF, with secondary systemic folate deficiency and CFD. MTHF supplementation may be recommended in these instances [32,58].

CFD has been associated with Rett syndrome due to X-linked MECP2 mutations and can be explained by the expression of non-functional FR pseudogenes, suspected to act as competitive non-functional inhibitors of FRα protein [59]. CFD has also been associated with a variant of Aicardi-Goutières syndrome, although the mechanism of CFD in this variant remains unclear [60].

Elevated concentrations of reactive oxygen species (ROS) predispose to increased catabolism of plasma MTHF pool, and down regulate both intracellular folate uptake and trans-cellular folate transport mechanisms by ROS mediated chemical reactions affecting the functional integrity of the FRα and *RFC1* proteins [31]. In addition, ROS interferes at different levels, resulting in dysfunction of various enzymes serving intermediary folate metabolism (methionine synthase), the enzyme neuronal tryptophan hydroxylase, and leading to oxidation of DNA [61,62,63,64,65].

## 5. Clinical Classification of Different CFD Syndromes Associated with FRα Autoantibodies

The first described CFD syndrome, now called infantile CFDS evolves to its full-blown picture during the first three years of life and starts from the age of 4–6 months of life with early symptoms of agitation, sleeping problems and unrest, which cannot be accounted for by other conditions presenting in early childhood (for example gastro-esophageal reflux, milk allergy, lactose intolerance). This is followed by neurodevelopmental delay, deceleration of head growth, hypotonia and ataxia, pyramidal signs in the lower limbs which can ascend to tetra-spasticity if left untreated and, in one-third of cases, seizures and dyskinesias. Progressive visual loss can occur from the age of three years and hearing loss from the age of six years onwards [1,3,17,18]. A minority of children with infantile-onset CFD syndrome develop autism [19,45]. Among half of the untreated CFD cases, MRI brain scans show delayed myelination with demyelinated subcortical and periventricular areas, cerebral and cerebellar atrophy [1]. Laboratory results are normal, including hematology, folate, B_12_, homocysteine, immunoglobulins, amino acids and screening for inborn errors of metabolism. However, a spinal tap will disclose low CSF folate levels, and, in the majority, serum tests positive for FRα autoantibodies [41]. It is important to suspect infantile CFD in these cases because early diagnosis and folinic acid treatment is essential to prevent further progression or neurologic deterioration. Children treated before the age of two years can recover, sometimes completely.

Compared to infantile-onset CFDS, the children manifesting low-IQ autism plus typical neurological features of the infantile-onset CFDS had equally low CSF MTHF levels and serum FRα autoantibody titers [45]. Apart from overt autism, we did not find any other differences between the two groups of infantile CFDS. Further investigations revealed that the parents of children with infantile-onset CFDS had absent FRα autoantibodies whereas FRα autoantibodies were present among either the mother, father or both parents of children with low-IQ autism and features of CFDS [41,43]. Because both sperm and oocytes depend on FRα mediated folate transport, autistic signs and symptoms might be related to the consequences of FRα autoimmunity in either one or both parents. Folate deficiency will have an impact on DNA within male sperm or oocytes, and consequently determine aberrant pathways deranging folate dependent metabolism, epigenetic programming of gene silencing/expression and neurodevelopment [66]. Moreover, the presence of FRα autoantibodies in mothers will further impair normal embryonic and fetal development and predisposes to neural tube defects (NTD) and infantile autism [67]. Other studies have confirmed that dietary folate deficiency during early pregnancy correlates with NTD and autism in the offspring [10,11,12,13].

A group of children with infantile autism, who had no neurological deficits, have been studied subsequently and revealed CSF MTHF levels which were not as low as compared to infantile CFD or low-IQ autism with neurological deficits, but were still low or within the low normal reference range with a statistical difference compared to the normal CSF folate reference range among healthy controls (Figure 3). Their FRα autoantibody titers are in the same range as those for children with infantile CFD or low-IQ autism with neurologic abnormalities. Compared to the almost complete absence of FRα autoimmunity in age-matched healthy control children and their parents, the parents of children with infantile autism had FRα autoimmunity in a significant proportion of the families [2,43]. Thus, in children with infantile autism, FRα autoimmunity can be acquired after birth by the child and can be absent or present in either one or both parents. For each family, eight different scenarios for the presence or absence of FRα autoimmunity for each trio (child, father, mother) can occur [43].

The occurrence of postnatal FRα antibodies in children between 1 and 2 years results in a spastic-ataxic CFD syndrome associated with learning deficits [68].

Between 2 and 5 years, FRα autoimmunity was observed in a low proportion of children suffering from attention deficit hyperkinetic disorder (ADHD), learning deficits and, sometimes, behavioral abnormalities. Neurologic examination shows discrete signs of gait ataxia and a positive Romberg test (personal observations).

During adolescence or adulthood, FRα autoimmunity can predispose to severe psychotic episodes and refractory schizophrenia characterized by marked FRα antibody titer fluctuations in cycles of 5–7 weeks which underlies either normal or lowered CSF folate values as a function of the FRα antibody titer at the time of the spinal tap. The first reported child suffered from catatonic schizophrenia [69]. Our studies reported among 20 patients with refractory schizophrenia positive FRα autoantibodies up to 85 % [46].

Other psychiatric conditions associated with FRα autoimmunity predisposing to CFD manifest as severe treatment-resistant major depression, in which among 9 out of 16 studied patients (56%), FRα autoantibodies were found. Folinic acid therapy in these cases shows some efficacy [2].

In patients with severe dystonia or Parkinsonism associated with psychiatric disorder, which do not respond to conventional treatment, a high index of suspicion of CFD due to FRα autoimmunity should be included in the work-up and differential diagnosis, as sporadic cases have been observed [70].

There is one report of an adult female suffering from dementia and myoclonus where a CFD syndrome was found [71]. No other reports on the different forms of dementia conditions were studied or reported in the literature.

In summary, the occurrence of a specific neurologic or psychiatric condition associated with CFD due to FRα autoimmunity depends on the age of onset when these FRα autoantibodies develop and on the presence or absence of parental FRα autoimmunity, particularly among families with children suffering from autism spectrum disorder (Figure 4).

## 6. Diagnostic Investigations

Based on the patient´s age, history and physical examination, a clinical diagnosis or classification of the condition can be established. Further neurological, psychiatric, and psychological testing will often be helpful at this initial stage to reach a firm diagnosis. If one of the CFD syndromes as described above is suspected, or there is a history of refractory dystonia, schizophrenia, or major depression with no improvement after conventional treatment, further specific investigations, such as brain imaging, EEG and neurophysiologic investigations, should be added [2,3].

If a CFD syndrome is suspected, the first diagnostic step is to determine a full blood count, serum and red blood cell folate, vitamin B_12_, homocysteine, lactate and other specific testing, as well as a full metabolic screening followed by a spinal tap to measure CSF glucose, protein and cells, and spinal fluid values for MTHF, intermediate and end-metabolites of mono-amines (dopamine, serotonin), the pterin metabolites, neopterin and biopterin, and in selected instances, lactate and amino-acids. In patients confirmed to suffer from a CFD syndrome, serum testing of FRα autoantibodies of the blocking and binding type should be tested on one or several occasions, since FRα antibodies may fluctuate over time. An initial sample can be negative, while testing positive on a later occasion [46]. Care should be taken to avoid taking vitamin supplements containing folates three days before FRα autoantibody testing. If FRα antibodies test negative, further investigations should exclude mitochondrial defects, *FOLR1* gene mutations, *MTHFR* gene mutations and, in selected instances, testing for the presence of oxidative stress biomarkers and the most common antioxidant scavenging vitamins as well as enzyme cofactors iron, copper, zinc, selenium and manganese for the antioxidant enzymes catalase, superoxide dismutase and glutathione peroxidase.

In this context, it should be stressed that in, for example, infantile CFD or autism, a spinal tap is not always necessary, and serum should be tested first to identify the presence of FRα autoantibodies, which are prevalent in these conditions and have been shown to correlate inversely with CSF MTHF levels [47].

A complete list of investigations necessary to identify the causes of systemic folate deficiency or CFDS can be adapted by the physician based accordingly on the most likely suspected diagnosis, since several conditions can be ruled out based on the patient´s individual history and physical examination (Figure 5).

## 7. Treatment

After diagnosis of CFD syndrome caused by FRα autoimmunity, treatment should be initiated immediately using high pharmacological doses of dl-folinic acid (daily dose of 0.5–1 mg/kg body weight dl-5-formyltetrahydrofolate which can be increased to 2 mg/kg/day with a maximum daily dose of 50 mg), levo-folinic acid (daily dose at 0.25–0.50 mg/kg body weight), or an equivalent dose of levo-5-methyl-tetrahydrofolate: the latter has not been approved for medical use in Europe [1,18]. Care should be taken to initiate folinic acid treatment slowly during the first month using half the finally required dose, since folate increases in the brain that are too abrupt will result in elevated production of tetrahydrobiopterin, and the neurotransmitters dopamine and serotonin, with consequent overstimulation of dopamine and serotonin synapses. It will take about 6 weeks before a new equilibrium will be established between augmented neurotransmitter production and the previously down regulated dopamine and serotonin receptors. Clinical symptoms of marked agitation and aggression occurs with too-abrupt folinic acid dose increases. If marked agitation or aggression still occur after the introduction of folinic acid treatment, a low dose of the neuroleptic agent risperidone can be used temporarily for 2 to 3 months.

Another therapeutic option, which can be combined with folinic acid treatment, is the introduction of a strict animal-milk-free diet using vegetal milk and products, so that exposure to soluble FR from animal-derived milk products can be avoided [47]. Dietary measures can be introduced as soon as diagnostic investigations (lumbar puncture, blood tests and serum FRα antibody testing) have been completed while awaiting test results for serum FRα antibodies. Following an animal-milk-free diet, FRα antibodies will drop significantly after 3–6 months. However, introduction of dietary measures is not always possible because particularly children with autism spectrum disorder will not accept or tolerate such a strict diet.

Infantile CFD syndrome due to FRα autoimmunity can sometimes manifest severe dyskinesia, frequently occurring seizures or, in rare circumstances, an intractable epileptic status. In such emergency situations, treatment with corticosteroids can be used to down regulate FRα autoimmunity prior to the emergence of a therapeutic effect from folinic acid and a milk-free diet. The use of intravenous gamma-globulin treatment has been tried but sufficient evidence is lacking to be certain on its short- and long-term beneficial effects.

Follow-ups of children with infantile CFD syndrome treated with folinic acid revealed that early treatment results in a better outcome. Diagnosis and treatment started before the age of two years can even cure signs and symptoms and lead to complete recovery [45]. For this reason, diagnosis should rely on a high index of suspicion by clinicians to identify and treat infantile-onset CFD syndrome as early as possible.

Likewise, children with infantile-onset autism associated with FRα autoimmunity should also be identified early to start folinic acid therapy and a diet as early as possible, to improve final prognosis. However, the simultaneous presence of FRα antibodies in the autistic child and either one parent impedes neurologic recovery and the resolution of core autistic signs and symptoms, following folinic acid therapy. Finally, the presence of FRα antibodies in both parents has a negative impact with poor outcome and prognosis despite folinic acid and dietary treatment [43]. Thus, if folinic acid therapy is not followed by the significant recovery of the child after four months, serum testing of FRα antibodies in both parents should be considered to elucidate the poor effect of treatment [43]. If treatment does not lead to any improvement, further metabolic and genetic testing should be considered as more than 1000 different genetic causes have been identified in autism spectrum disorders [72] (see also SFARI gene database). Additional testing for inborn errors of metabolism and genetic causes of autism spectrum disorder should be considered early in the initial diagnostic process if dysmorphic features are found in the presence of organ malformations, parental consanguinity, or other affected first-degree family members. The use of new genetic techniques (array CGH and WES analysis) can be helpful to reach a diagnosis for this group of genetically determined disorders [72].

CFD syndrome, due to loss of function *FOLR1* mutations, results in very low CSF MTHF values below 10 nmol/L; therefore, higher daily doses up to 5–7 mg/kg are required to restore cerebral folate and reduce clinical manifestations [73]. In hereditary folate malabsorption caused by *PCFT* gene mutations, similar high folinic acid intramuscular doses will be necessary to correct low plasma and CSF folates [16].

CFD syndrome associated with mitochondrial defects, such as Alpers disease or Kearns–Sayre syndrome, are responsive to folinic acid to some extent with variable outcomes, or at least the stabilization of the condition [48,49,50,51]. The addition of antioxidants can further help to combat oxidative stress but evidence for significant improvement is still lacking [51].

In many children suffering from autism, selective feeding disturbances, malnutrition and lack of vitamins and trace elements can be found so that systematic screening for these parameters should be included in the diagnostic work-up. A number of cases with infantile autism with absent FRα autoantibodies in the child and parents have been recognized, in whom a deficiency of vitamin B_12_, multiple vitamins, metals and trace elements was present [62]. Because the vitamins C, E and gamma-tocopherol, co-enzyme Q10 as well as iron, copper, zinc, manganese and selenium function as radical scavengers or cofactors of antioxidant enzymes, reactive oxygen species will increase and interfere with intermediary folate metabolism and the folate transport mechanisms provided by FRα and *RFC1*. Thus, it is mandatary to supply these deficient parameters in the context of successful treatment and management of CFD syndromes [62,74].

Vitamin D deficiency occurs commonly and will reduce *RFC1* gene expression, which provides the alternative route at high plasma folate concentrations to enable reduced folates to pass the blood–brain barriers. Therefore, it is essential to correct vitamin D deficiency in combination with folinic acid treatment for CFD syndromes [75].

Vitamin riboflavin supplements are required at high doses to increase MTHFR enzyme activity in cases where a homozygous C677T mutation or compound heterozygous C677T and A1298C mutations will reduce MTHFR enzyme activity by 50–60% and increase homocysteine levels. After folinic acid (5-formyl-THF) has passed through two enzymatic steps to form 5,10-methylene-THF, the riboflavin-dependent enzyme MTHFR will reduce 5,10-methylene-THF to 5-methyl-THF. In the advent of these common mutations with 50 to 60 % diminished MTHFR enzyme activity, the plasma folate pool contains diminished MTHF concentrations and impairs folate passage to the brain because MTHF is the sole reduced folate form to cross the blood–brain barrier. In these instances, part of the folinic acid dose should be replaced by an equivalent levo-methyl-THF dose [76,77,78,79].

Patients with refractory psychotic or schizophrenic syndromes are often found to suffer from fluctuating FRα autoantibody titres where phases with high titres lead to CFD with downregulation of tetrahydrobiopterin production and consequent reduced serotonin, dopamine, and NO production, whereas lowering of FRα antibody titres have been associated with a boost of tetrahydrobiopterin production and elevated biogenic monoamine stimulation. A link with the clinical course has been suspected which can explain the periods with negative or positive symptoms [2,46].

Among patients with major depression, FRα autoimmunity can be detected and should be followed by folinic acid administration at high doses [2].

## 8. Perspective

One of the important aspects in the clinical management of infantile-onset CFDS, autism with neurologic deficits or infantile autism requires a high index of awareness, particularly in young children, to diagnose these conditions as early as possible since early folinic acid treatment for FRα autoimmunity carrying a high prevalence in these conditions will considerably improve outcome [74].

In addition, prognosis becomes less favorable if one or both parents are positive for these FRα autoantibodies. Most parents carrying antibodies remain asymptomatic [44]. For future pregnancies among affected families, parents carrying these autoantibodies should receive folinic acid at low doses up to daily doses of 3.75 mg levo-folinic acid, which should be started three months before conception and throughout pregnancy in mothers positive for FRα autoimmunity. Preliminary results from three families, where the first-born child suffered from infantile autism and parents tested positive for FRα autoantibodies, showed that folinic acid treatment of parents resulted in an offspring of five completely healthy siblings (Figure 6). Before a second pregnancy is planned, genetic counselling should always exclude the possibility of genetically determined causes of autism.

In children with CFD syndromes and rare inborn errors of metabolism, genetic causes affecting the function of FOLR1, CIC and common MTHFR mutations, as well as mitochondrial defects and abundant reactive oxygen species should be considered. For this purpose, the proposed diagnostic algorithm including spinal fluid analysis should be considered. Early diagnosis and treatment are key to a more favorable outcome. Screening for maternal and cord blood FRαAb, and subsequently monitoring the child for FRαAb at 6, 12 and 24 months should be considered. Prevention of CFDS and ASD in the offspring is a distinct possibility if potential parents are screened for FRαAb and genes involved and treated with folinic acid prior to conceiving. These suggestions warrant appropriate clinical trials to prove the success of early intervention.

We have started to analyze children with infantile CFD using whole exon sequence screening among patients and their parents. Results of possible hereditary factors underlying these neurodevelopmental disorders and FRα autoimmunity are pending.

## Figures and Tables

**Figure 1 nutrients-14-03096-f001:**
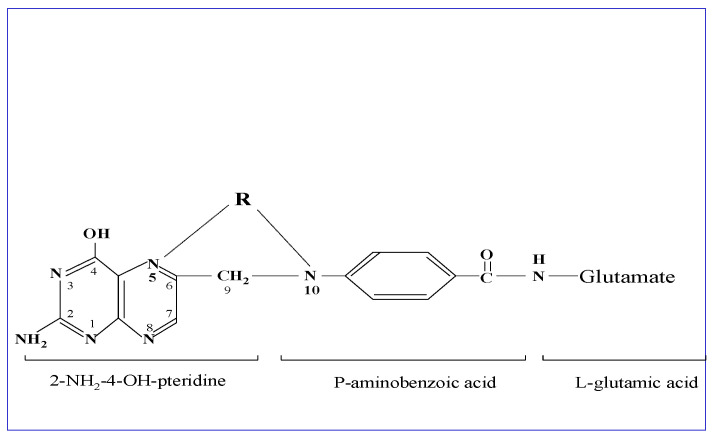
Structural formula of folic acid. This is composed of an unreduced 2-amino-4-hydroxypteridine molecule linked through methylene (C6-position) to p-aminobenzoylmono-glutamate, while R represents the one-carbon group at different oxidation states in the form of methyl-, methenyl-, methylene-, or formyl-, attached to the N5- and/or N-10 positions as shown in the drawing. The metabolically active form tetrahydrofolate is reduced at the 5,6,7,8 positions of the pteridine ring.

**Figure 2 nutrients-14-03096-f002:**
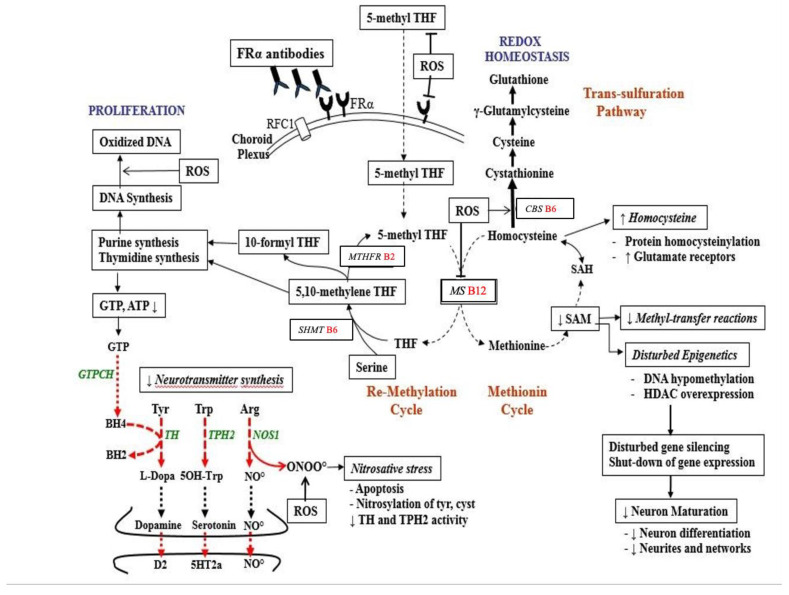
Pathways of folate metabolism after FRα mediated flux of 5-methyl-tetrahydrofolate (5-methyl THF) across the choroid plexus into the CSF and then to the brain. Inside neurons, most methyl-THF will be stored while a proportion actively participates in metabolism. Methyl-THF transfers its methyl group to the B12-dependent methionine synthase (MS) enzyme, converting homocysteine to methionine. In the methionine cycle methionine is the precursor of the universal methyl-donor SAM, used in more than 100 methylation reactions. In the remethylation cycle, tetrahydrofolate (THF) receives a mono-carbon group from serine and is converted to 5,10-methylene THF. Part of 5,10-methylene THF is reduced by methylene-THF reductase (MTHFR B2) to 5-methyl-THF, while another portion is used to produce thymidine or is converted to 10-formyl THF, needed for purine synthesis. On the left side of the picture, the purine metabolite GTP serves as substrate for the GTP-cyclohydrolase I enzyme (GTPCH) to produce tetrahydrobiopterin (BH4), which is the co-factor for enzymes producing dopamine, serotonin and NO. At the upper right of the figure the transsulfuration pathway is shown. It converts accumulated homocysteine to the antioxidant glutathione to ameliorate oxidative stress. Abbreviations: BH4: tetrahydrobiopterin; D2: dopamine-2 receptor; GTP: Guanosine Tri Phosphate; NO: Nitric oxide; NOS: Nitric oxide synthase; *RFC1*: Reduced Folate Carrier-1; ROS: reactive oxygen species; TH: Tyrosine hydroxylase; TPH2: neuronal Tryptophane hydroxylase; 5HT2a: serotonin 2A receptor; SAM: S-adenosyl methionine; SAH: S-adenosyl homocysteine.

**Figure 3 nutrients-14-03096-f003:**
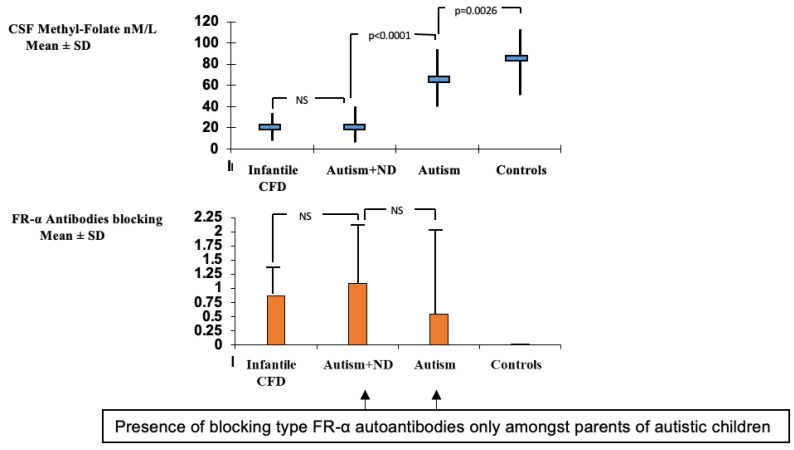
Comparison of CSF methyl-folate levels and FRα autoantibodies in infantile CFD, autism with neurologic deficits (Autism + ND), infantile autism and healthy controls.

**Figure 4 nutrients-14-03096-f004:**
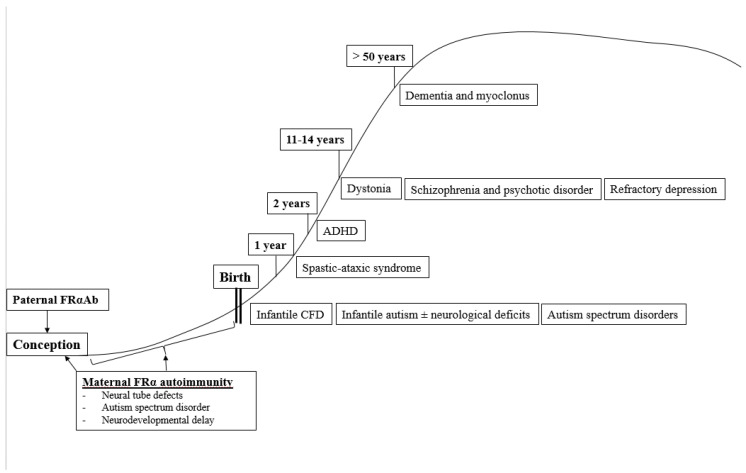
Age-dependent classification of CFDS.

**Figure 5 nutrients-14-03096-f005:**
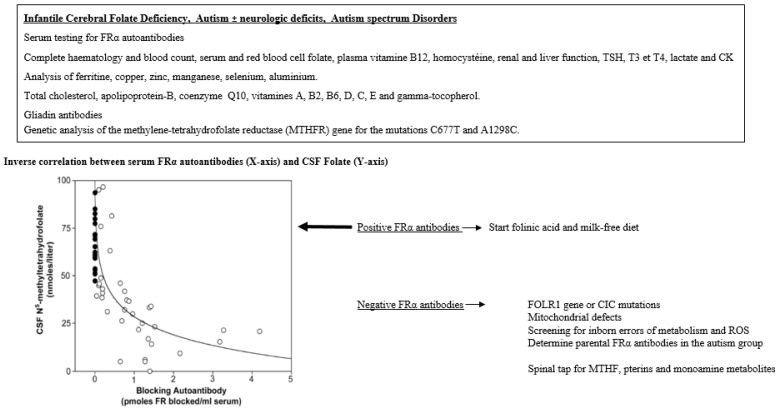
Diagnostic alghorhytm for infantile CFD and autistic syndromes.

**Figure 6 nutrients-14-03096-f006:**
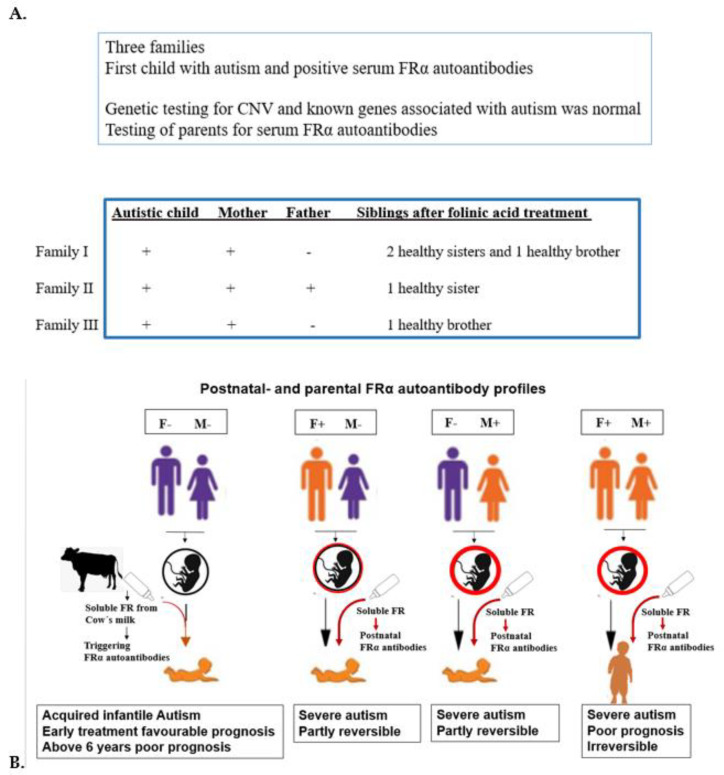
(**A**). Genetic counseling FRαAb testing for first child with autism and parents to prevent the reoccurrence of autism in further offspring. (**B**). Various situations with positive FRα autoanti- bodies in child and parents.

**Table 1 nutrients-14-03096-t001:** Five potential mechanisms underlying CFD.

**I. Reduced transport of folates across the blood–brain barrier into CSF and across the ependymal barrier into the brain**
* 1. Disorders of Folate Receptor alpha (FR* *α).*
- Presence in serum of blocking and/or binding autoantibodies against folate receptor (FRα) (infantile-onset CFD, spastic-ataxic CFD, dystonia CFD, Rett syndrome, Aicardi-Goutières variant, autism spectrum disorders, part of attention deficit-hyperkinetic syndrome, schizophrenia
- Genetic defects leading to loss of function of the FRα, dysregulation of its expression or attachment processes to the external cellular membrane
- Signal transduction disorders affecting the regulation of FRα-mediated endocytosis
- Disorders affecting membrane fluidity and composition (Smith-Lemli-Opitz syndrome)
* 2. Disorders of energy metabolism and ATP production*
- Mitochondrial disorders (Kearns Sayre syndrome, Complex I-V deficiencies, mitochondrial DNA depletion Alpers syndrome)
- Glucose transporter deficiency (GLUT 1-deficiency)
- Any disorder or condition with intracellular energy failure.
* 3. Conditions damaging the transport functions of brain-endothelial vessel walls and choroid epithelial cells*
- Intracranial choroid plexus bleeding (premature infants)
- Xanthogranulomatous lesion of the choroid plexus
- Infectious, immunologic agents and traumatic lesions
- Elevated reactive oxygen species or peroxynitrite with consequent damage to membranes and folate transporting proteins (FRα, Reduced Folate Carrier)
**II. Decreased folate storage due to depletion of intracellular folyl-polyglutamate pool.**
**III. Increased utilization of reduced folates in the brain**
* 1. Hereditary conditions*
- Aromatic amino acid decarboxylase deficiency
- Dihydropteridine Reductase deficiency
* 2. Iatrogenic conditions*
- Inhibitors of the enzyme aromatic amino acid decarboxylase (use of carbidopa or benserazide combined with L-Dopa in Parkinson disease)
* 3. Infectious, parainfectious and immune-mediated conditions*
- Subacute Sclerosing Pan Encephalitis (SSPE)
- Reactivated herpes infections
- Rasmussen encephalitis
**IV. Increased catabolism of reduced folates within the nervous system**
* 1. Conditions associated with oxidative and/or nitrosative stress*
- Increased production of reactive oxygen species (mitochondrial disorders, inflammatory processes) or failure of antioxidant enzymes and radical scavenging defenses
+ Glutathione Peroxidase deficiency associated with selenium deficiency
+ Extra-cellular and intracellular Superoxide Dismutase deficiency secondary to intracellular and/or extracellular manganese deficiency
+ Ubichinone-10 deficiency states
+ All conditions associated with vitamin C and E deficiencies
* 2. Inflammatory processes*
**V. Conditions affecting folate metabolism in the brain**
* 1. Hereditary conditions*
Enzyme deficiencies
- Methylenetetrahydrofolate reductase deficiency
- Serine hydroxymethyltransferase deficiency
- Dihydrofolate reductase deficiency
- Dihydropteridine reductase deficiency
Depletion of methyl-donor pool glycine, serine, and histidine
- Disorders of serine synthesis (3-phosphoglycerate dehydrogenase deficiency)
- Glutamate Formiminotransferase deficiency
* 2. Hyper homocysteinemia due to combined deficiencies of vitamin B2, B6, B9 and B_12_*

**Table 2 nutrients-14-03096-t002:** Overview of systemic folate depletion and conditions of CFD associated with isolated 5MTHF loss in the CNS.

Classification of Condition	Underlying Mechanism
**Systemic depletion **	
**Malnutrition**	Folate deficient diet or food deprivation
**Malabsorption**	Decreased folate absorption in the jejenum
- Coeliac disease	Gluten enteropathy
- Chron´s and jejunal diseases	Affects folate absorption
**Antifolate agents**	
- Chemotherapy (MTX, 5FU, Pemetrexate)	Blocks *RFC1* carrier; inhibits dihydrofolate reductase
- Tuberculostatic drug	Mechanism unknown
- Sulfonamides	Analogs of para-aminobenzoic acid, interfering with tetrahydrofolate synthesis in sensitive bacteria
- Anticonvulsant drugs	Interfering with cellular folate uptake
- Carbidopa	Inhibits aromatic amino acid decarboxylase and consequently, SAM and MTHF overconsumption
**Congenital folate malabsorption**	Hereditary factor involving the *PCFT* gene
**Inborn errors of metabolism**	
- Methylenetetrahydrofolate reductase def.	Depletion of enzymatic product MTHF
- Glutamate formiminotransferase deficiency	Defective histidine derived one-carbon transfer to tetrahydrofolate
- Dihydrofolate reductase deficiency	Reduction of the reduced folate pool
- Methylenetetrahydrofolate dehydrogenase I deficiency.	Reduction of 5-formyl and 10-formyltetrahydrofolate pool
**Cerebral Folate Deficiency **	
**Infantile-onset CFD**	Serum FRα autoantibodies of the blocking and/or binding type
**Mitochondrial encephalopathies**	Decreased active folate transport at choroid plexus
**Kearns-Sayre syndrome**	Decreased active folate transport at choroid plexus
**Alper’s disease**	Decreased active folate transport at choroid plexus
**CFDS due to FRα dysfunction**	FOLR1 gene defects, de novo mutation of Capicua transcriptional repressor (CIC) gene
**One-carbon pool deficiencies in the CNS**	
- 3-Phosphoglycerate dehydrogenase deficiency	Defective serine synthesis affecting one-carbon pool
**Inborn errors of metabolism**	
- Methenyltetrahydrofolate synthase deficiency	Defective conversion of 5-formyl-THF to 5–10-methenyl-THF
- Dihydropteridine reductase deficiency	Diminished conversion of dihydrofolate to THF
- Aromatic L-amino acid decarboxylase def.	Overconsumption of 5MTHF and SAM
**Rett syndrome**	FRα pseudo gene expression due to MECP2 defect
**Variant of Aicardi-Goutières syndrome**	Variable CFD of unknown origin
**Reactive Oxygen Species**	MTHF instability and dysfunction of FRα and RFC1

## Data Availability

Not Applicable.

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
