# Peer review of "Cerebral Folate Deficiency Syndrome: Early Diagnosis, Intervention and Treatment Strategies"

_nutrients, 2022, doi:10.3390/nu14153096_

Round 1
Reviewer 1 Report
It is very interesting and significant review work on CFDS. It is well known that folate is key micronutrient for development of fetus. When maternal deficiency in folate during pregnancy, the risk of adverse birth outcomes such as NTDs or CHDs in offspring. CFDS could be another. There are some comments as following:
1. Whether are there some information on epidemiology of CFDS now? if yes, it would be better to add them, which will be helpful to understand the epidemiological features of DFDS.
2. In the part of “Causes of cerebral folate deficiency”, authors said that “systemic folate deficiency occurs frequently and is due to malnutrition with additional common deficiencies of iodine, iron, zinc and essential vitamins like vitamin A, D and B12”. It sounds little confusing. Dietary folate intake is key for maintaining body folate, especially for susceptible population as pregnant women. Maybe dietary malnutrition or inadequate dietary intake of foods rich in folate or vitamin Bs are better.
3. Prevention of CFDS is possible according to current progress? saying something more about this issue.
Author Response
It is very interesting and significant review work on CFDS. It is well known that folate is key micronutrient for development of fetus. When maternal deficiency in folate during pregnancy, the risk of adverse birth outcomes such as NTDs or CHDs in offspring. CFDS could be another. There are some comments as following:
- Whether are there some information on epidemiology of CFDS now? if yes, it would be better to add them, which will be helpful to understand the epidemiological features of DFDS. Unfortunately, epidemiologic data on this disorder is scarce currently.
- In the part of “Causes of cerebral folate deficiency”, authors said that “systemic folate deficiency occurs frequently and is due to malnutrition with additional common deficiencies of iodine, iron, zinc and essential vitamins like vitamin A, D and B12”. It sounds little confusing. Dietary folate intake is key for maintaining body folate, especially for susceptible population as pregnant women. Maybe dietary malnutrition or inadequate dietary intake of foods rich in folate or vitamin Bs are better. The statement has been modified to refer only to folate deficiency.
- Prevention of CFDS is possible according to current progress? saying something more about this issue. Prevention of birth defects and CFDS are an objective going forward. A statement has been added in the final paragraph.
Reviewer 2 Report
This is a comprehensive review. There are some minor revisions that need to be completed. A number of figures have words underlined in red, is there a purpose for this? If not, can these lines be removed?
Could table 1 and 2 be revised so that they are more clear?
I think that the manuscript would benefit from a conclusion.
Author Response
Comments and Suggestions for Authors
This is a comprehensive review. There are some minor revisions that need to be completed. A number of figures have words underlined in red, is there a purpose for this? If not, can these lines be removed? This is now corrected.
Could table 1 and 2 be revised so that they are more clear? We have attempted to simplify the tables
I think that the manuscript would benefit from a conclusion. We have extended the concluding paragraph to include additional statements.